# Global soil profiles indicate depth-dependent soil carbon losses under a warmer climate

Mingming Wang [1,8], Xiaowei Guo [1,8], Shuai Zhang [1], Liujun Xiao [1], Umakant Mishra[2], Yuanhe Yang [3], Biao Zhu [4], Guocheng Wang [5], Xiali Mao[1], Tian Qian[1], Tong Jiang[1], Zhou Shi [1,6,7] & Zhongkui Luo [1,6,7] ✉

Soil organic carbon (SOC) changes under future climate warming are difficult to quantify in situ. Here we apply an innovative approach combining space-for-time substitution with meta-analysis to SOC measurements in 113,013 soil profiles across the globe to estimate the effect of future climate warming on steady-state SOC stocks. We find that SOC stock will reduce by $6.0 \pm 1.6\%$ (mean±95% confidence interval), $4.8 \pm 2.3\%$ and $1.3 \pm 4.0\%$ at 0–0.3, 0.3–1 and 1–2 m soil depths, respectively, under 1 °C air warming, with additional 4.2%, 2.2% and 1.4% losses per every additional 1 °C warming, respectively. The largest proportional SOC losses occur in boreal forests. Existing SOC level is the predominant determinant of the spatial variability of SOC changes with higher percentage losses in SOC-rich soils. Our work demonstrates that warming induces more proportional SOC losses in topsoil than in subsoil, particularly from high-latitudinal SOC-rich systems.

Under future warmer climate, whether soils are a sink or source of atmospheric $CO_2$ depends on the responses of both plant growth (which determines carbon inputs to soil) and soil organic carbon (SOC) decomposition (which determines carbon outputs to the atmosphere)[1,2]. Due to the complexity of climate-plant-soil interactions, biophysical Earth system models integrating plant growth and soil biogeochemical processes are commonly used to predict carbon cycle-climate warming feedbacks[3,4]. Nevertheless, model-derived predictions contain large uncertainties and have been widely debated due to our limited ability to model the high heterogeneity of carbon stabilisation and destabilization processes and their interaction with plants[5,6]. In addition, direct long-term measurements of soil carbon inputs and outputs as impacted by warming are lacking[7] and our understanding of underlying mechanisms is also insufficient[8], hindering reliable prediction of net SOC changes at the time scale of SOC turnover which may be centuries or millennia[9–12].

Climate warming (or any directional climate changes) and its influences are chronic, non-linear, and need decades or centuries to manifest[13–15]. A series of ecosystem processes (e.g., vegetation composition and structure, soil microbial community and functioning) may gradually transform or adapt to such chronic warming[16–19]. However, experimental manipulation of temperature regardless of in or ex situ is usually implemented by a step change of temperature and/or warming a specific component of the ecosystem (i.e., partial warming, e.g., only soil is warmed but not forest canopy in experiments conducted in forest ecosystems[20,21]) and relatively short-duration (days/months in laboratory incubations up to years/decades in field warming experiments)[22,23]. The short-term nature of these manipulative experiments makes it impossible to capture gradual shifts of both above- and below-ground processes (e.g., vegetation transformation) and the relevant consequences on long-term net SOC balance when the soil at last reaches a new steady state under the manipulative

[1]College of Environmental and Resource Sciences, Zhejiang University, 310058 Hangzhou, China. [2]Computational Biology & Biophysics, Sandia National Laboratories, Livermore, CA, USA. [3]State Key Laboratory of Vegetation and Environmental Change, Institute of Botany, Chinese Academy of Sciences, 100093 Beijing, China. [4]Institute of Ecology, College of Urban and Environmental Sciences, and Key Laboratory for Earth Surface Processes of the Ministry of Education, Peking University, 100871 Beijing, China. [5]State Key Laboratory of Atmospheric Boundary Layer Physics and Atmospheric Chemistry, Institute of Atmospheric Physics, Chinese Academy of Sciences, 100029 Beijing, China. [6]Academy of Ecological Civilization, Zhejiang University, 310058 Hangzhou, China. [7]Key Laboratory of Environment Remediation and Ecological Health, Ministry of Education, Zhejiang University, 310058 Hangzhou, China. [8]These authors contributed equally: Mingming Wang, Xiaowei Guo. ✉e-mail: luozk@zju.edu.cn

condition. Indeed, current results on net SOC balance under warming are inconclusive and uncertain depending on experimental conditions, edaphic properties, baseline climate, and ecosystem type[24–26]. A five-year whole-soil warming experiment in a temperate forest found opposite responses of SOC stock to warming in topsoil and subsoil[27]. As it is challenging via relatively short-term warming experiments, if not impossible, to obtain direct large-scale observations to holistically capture long-term responses of ecosystems[28], innovative approaches are needed to address the inconsistency and uncertainty of whole-soil-profile net SOC balance under future warmer climate.

In this study, we take advantage of a global data set of SOC measurements in 113,013 soil profiles across the globe[29] which includes 2,703 soil profiles in the northern hemisphere permafrost region[30] (Supplementary Fig. 1) to assess the responses of both SOC content ($SOC_c$, g C kg$^{-1}$ soil) and stock ($SOC_s$, Mg C ha$^{-1}$) to climate warming, using a hybrid approach combining space-for-time substitution[31] with meta-analytic techniques (Fig. 1). First, the 113,013 soil profiles were sorted by mean annual temperature (MAT) at the profile locations and divided into classes distinguished by MAT. Depending on the warming level of interest (i.e., 1, 2, 3, 4, and 5 °C in this study), an "ambient" and a "warm" class were selected. That is, MAT in the "warm" class must be certain degrees (i.e., 1, 2, 3, 4 or 5 °C) higher than that in the "ambient" class. Considering the potential effects of precipitation including its seasonality, landform and soil type, each class was further divided into groups distinguished by mean annual precipitation, precipitation seasonality, landform and soil type. Then, meta-analytic techniques were applied to the two groups (an "ambient" group vs a "warm" group) that share the same precipitation, landform and soil type to estimate the percentage response of $SOC_c$ as well as $SOC_s$ to warming (i.e., the difference of MAT between the "ambient" and "warm" groups). Our approach implicitly adopts the steady state assumption. That is, soils and thus SOC under current climate represented by the "ambient" group are at steady state, and the soils will finally reach a new steady state under future warmer climate represented by the "warm" group. So, the estimated percentage response of SOC represent the difference between the two steady states.

## Results and discussion
### Depth-dependent responses of SOC to warming
The meta-analysis results indicate that $SOC_s$ in the 0–0.3 m soil is reduced by 6.0 ± 1.6% (mean ± 95% confidence interval) under 1 °C warming (Fig. 2a). In the 0.3–1 m and 1–2 m soil layer depths, the loss is reduced to 4.8 ± 2.3% and 1.3 ± 4.0%, respectively (Fig. 2a). With increasing warming level, regression indicates that $SOC_s$ losses will increase at a rate of 4.2%, 2.2% and 1.4% per additional 1 °C warming in the three layers, respectively (Fig. 2a). Similar responses are also observed for $SOC_c$ (Fig. 2b). To address the effect of the steady state assumption on the results, we conducted two sensitivity analyses (Methods section). First, soil profiles from croplands were excluded from the meta-analysis as cropland soils have a high probability of not being at steady state. For both $SOC_s$ and $SOC_c$, estimated responses are similar to those including cropland soil profiles (Supplementary Fig. 2). Second, groups with <20 soil profiles were excluded from the meta-analysis. This allows the assessed pairs (which include an ambient and a warm group) to cover a higher diversity of soil conditions such as land history and future land cover/use, diluting the effect of non-steady state soil on the estimates. The results indicate that uncertainty in the estimates is increased due to the decrease in sample size, but the general net SOC loss does not change (Supplementary Fig. 2).

These results demonstrate that on average global soils will be a source of carbon to the atmosphere under future warmer climate (i.e., positive soil carbon loss-climate warming feedbacks). However, SOC in deeper layers show smaller losses in general (Fig. 2 and Supplementary Figs. 3 and 4). This may be attributed to the inhibition of lower oxygen availability and less high-quality carbon substrates to SOC decomposition in deeper soil depths[32]. If SOC decomposition and turnover are predominantly determined by other environmental constrains rather than temperature, temperature sensitivity of SOC decomposition would be largely attenuated[1]. In addition, under a warmer climate, given otherwise similar environmental conditions, plants may invest more carbon to growth deeper roots to acquire water, compensating SOC losses in subsoil[33]. Another important reason would be that the magnitude of soil temperature change may not keep the same pace

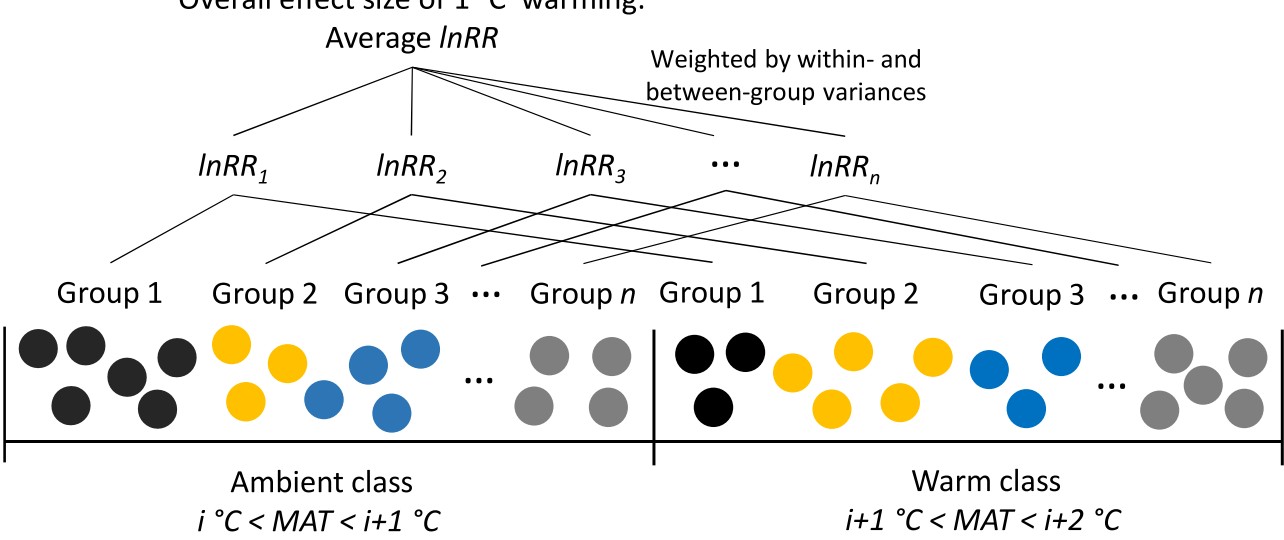

**Fig. 1 | Schematic representation of the approach used to quantify the response of soil organic carbon (SOC) to warming.** Each dot represents one soil profile. Dots with the same colour indicate that they share the same mean annual precipitation, precipitation seasonality, landform and soil type, and are grouped into classes distinguished by mean annual temperature (MAT). Depending on the interest of warming level (e.g., 1 °C in this schematic example), two classes (i.e., an ambient class and a warm class) are selected and SOC measurements (content and stock) from soil profiles belong to the same group are compared between ambient and warm classes (e.g., group 1 in ambient class vs group 1 in warm class). Meta-analytic techniques are applied to calculate the effect size for each comparison (i.e., the log response ratio lnRR), and a weighted average effect size by the inverse of the sum of within- and between-group variances is estimated.

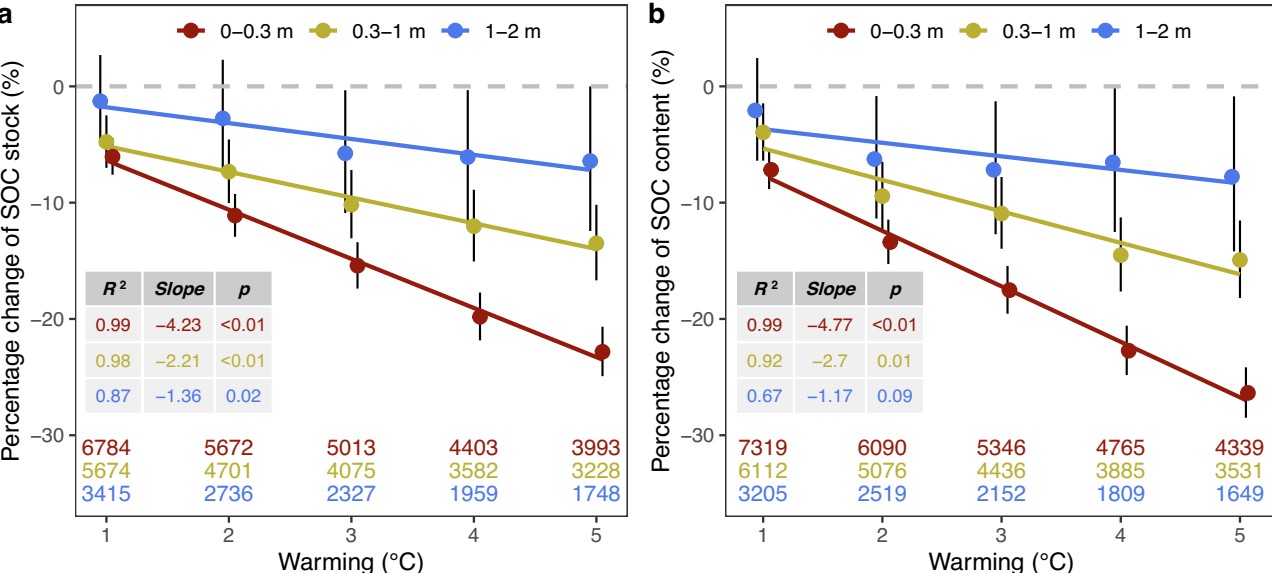

**Fig. 2 | The resoponse of soil organic carbon (SOC) to warming.** Global percentage response of SOC stock (a) and content (b) to warming. Sample size (i.e., the number of soil profiles used to estimate the response) is shown for each soil layer depth (i.e., 0–0.3, 0.3–1 and 1–2 m) under each warming level (i.e., 1, 2, 3, 4, 5 °C warming). Black vertical bars show the 95% confidence interval. Solid lines show the linear regression line between response and warming level in three soil layer depths, and inset table shows statistics for the regression including the determination coefficient ($R^2$), regression slope and $p$ value. Values for warming levels are jittered to make the points more distinct.

with air temperature change due to energy dissipation with heat conduction and diffusion through the soil profile. In deep layers, soil temperature may be also strongly modulated by belowground geological characteristics (e.g., groundwater table, depth of bedrock). The consequence of such discrepancy between air and soil temperature changes on the estimation of whole-soil SOC dynamics in response to climate warming is rarely assessed[34]. If SOC in deeper layers do not have much weaker temperature sensitivity as observed by a series of field and laboratory experiments[21,35,36], similar proportional net SOC changes under the same soil temperature changes should be expected. Our results of smaller percentage SOC losses in deeper layers suggest that surface air warming may induce depth-dependent soil temperature increases, which should be carefully assessed. As the "ambient" and "warm" groups used in the meta-analysis were selected by surface air temperature, the space-for-time substitution approach used in this study would partially capture the difference between soil and air temperature changes.

**Controls over SOC responses**
The responses of SOC to warming vary significantly across biome types (Supplementary Figs. 3 and 4 and Supplementary Tables 1 and 2). In most biomes, $SOC_s$ show significant negative responses in all three soil layer depths. In tundra systems, however, contrary to the expectation of negative response of SOC to warming[17,37,38], $SOC_s$ are increased irrespective of soil depth and warming level, albeit the increase is statistically insignificant (Supplementary Fig. 3). There is evidence that warming may substantially increase carbon inputs to the soil through enhanced vegetation growth in cold systems, thus offsetting SOC losses induced by increased decomposition[26,39,40]. Another intriguing finding is that SOC in different biomes do not show the same response to increasing warming level. For example, $SOC_s$ reduction in the 0−0.3 m soil is 10% and 34% under 1 °C and 5 °C warming, respectively, in temperate forests (i.e., 24% absolute change), but this absolute change under the two temperatures is only 10% in tropical/subtropical forests (Supplementary Fig. 3). In tundra systems, the response of $SOC_s$ in the top 0.3 m soil is relatively stable under all warming levels. These results highlight biome-specific sensitivity of SOC dynamics to warming.

The responses of SOC to warming are also regulated by precipitation including its seasonal pattern, but are less influenced by soil type and landform (Supplementary Figs. 3 and 4 and Supplementary Tables 1 and 2). Here we note that the soil type assessed is based on the world map of soil orders of the USDA soil classification system. To verify the effect of soil type (Supplementary Tables 1 and 2), we also used FAO and WRB soil groups in the estimation (Methods section). The results indicate that using soil types from the three soil classification systems generates similar results on the response of SOC to warming (Supplementary Fig. 5). One may argue that the soil orders/groups used are too coarse to reflect soil properties. We conducted an additional assessment based on the 63 soil suborders in the USDA system. This also does not change the estimation of the responses of both $SOC_s$ and $SOC_c$ to warming (Supplementary Fig. 5). Soil properties are recognised to be important for SOC stabilisation and storage[8,41,42]. It is reasonable to expect that soil properties may also shape the response of SOC to warming. Soil types assessed here can to some extent reflect common soil physiochemical conditions, but our assessment does not support that soil type plays a significant role in regulating overall SOC balance under climate warming at the global scale. At finer scales, we acknowledge that heterogeneous soil properties may be important.

To assess the relative importance of biome type, precipitation pattern, soil type and landform, together with a suite of other environmental variables, we trained a random forest model adapted for meta-analysis (meta-forest) which considers interactions among the variables and potential nonlinearity (Methods section). The meta-forest model shows good predictive power for SOC changes induced by warming (tenfold cross-validated $R^2 = 0.69$ for $SOC_s$ changes and 0.64 for $SOC_c$ changes, Fig. 3). For $SOC_s$, existing $SOC_s$ is the most important variable influencing its percentage changes under warming (Fig. 3a), and high SOC level is associated with higher SOC losses in all layer depths (Supplementary Fig. 6). Following the existing SOC stock, baseline mean annual precipitation and temperature are the two most important variables. This result may imply the important role of trade-offs between soil moisture and thermal regimes in controlling the response of SOC to warming[43,44]. Similar variable importance is also found for the response of $SOC_c$ (Fig. 3b). These results suggest the vital

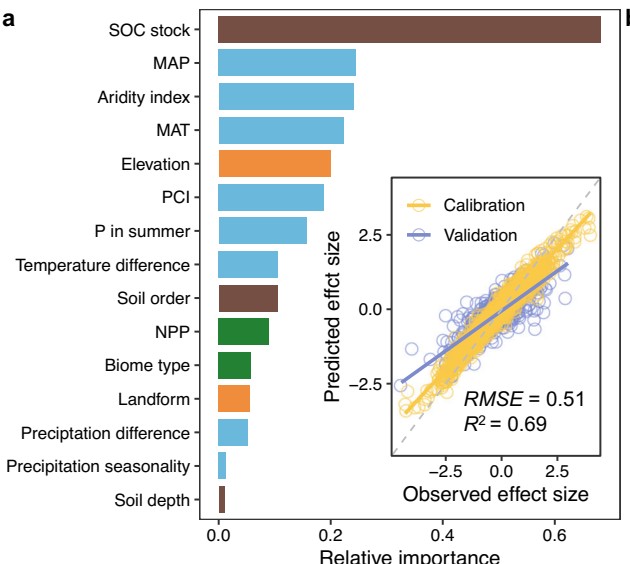
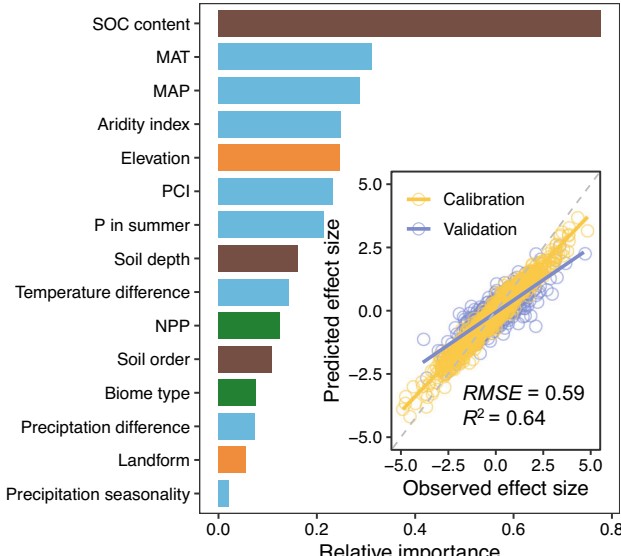

**Fig. 3 | The relative importance of environmental variables in influencing the response of soil organic carbon (SOC) to warming. a**, **b** the results for SOC stock and content, respectively. The relative importance is identified by a meta-forest model driven by the listed variables to predict the response of SOC stock and content to warming. MAP, mean annual precipitation; MAT, mean annual temperature; PCI, precipitation concentration index; P in summer, the fraction of precipitation in summer to MAP; NPP, net primary production. Insets show the performance of the meta-forest model with RMSE and $R^2$ show the rooted mean squared error and determination coefficients, respectively. Brown bars, soil-related variables; blue bars, climate-related variables; grey bars, topography-related variables; green bars, vegetation-related variables.

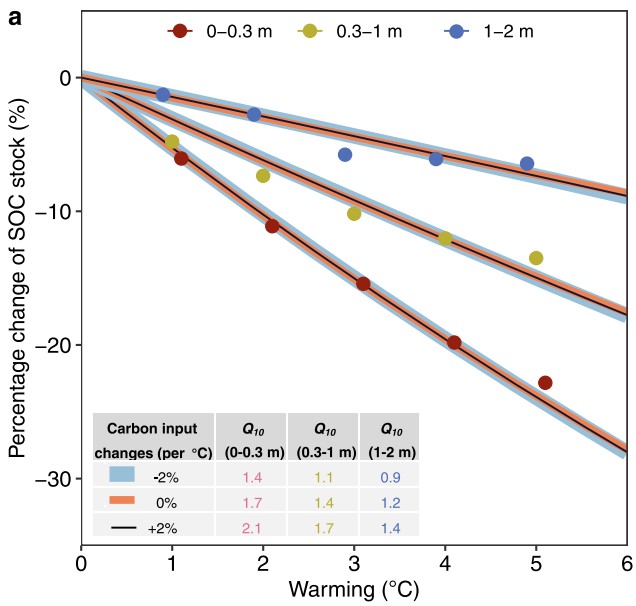
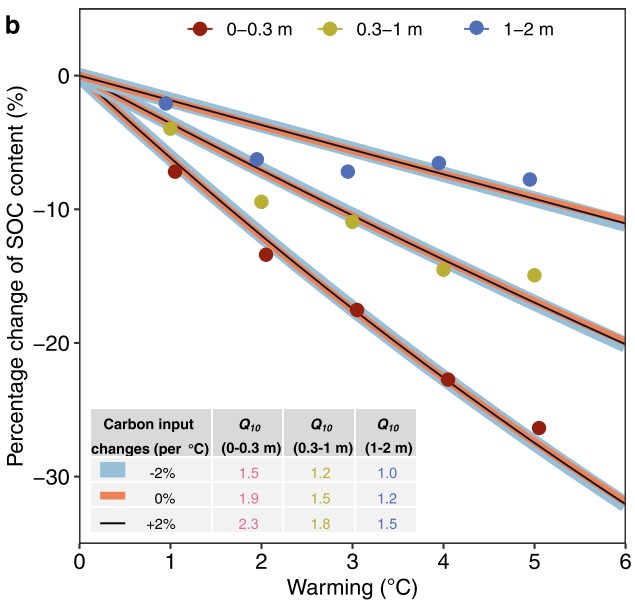

**Fig. 4 | Comparison with soil carbon model-estimated responses of soil organic carbon (SOC) to warming. a** SOC stock, **b** SOC content. Points are averages estimated using the space-for-time substitution approach developed in this study. Lines are predictions of a one-pool carbon model using optimised $Q_{10}$ values, with different line width indicating carbon input change scenarios under warming. The optimised $Q_{10}$ values are shown in the inset table.

role of baseline climatic conditions in regulating SOC dynamics in response to warming. The importance of aridity index and precipitation centralization index (PCI) further demonstrates this (Fig. 3).

## Comparison with predictions of soil carbon models

We compared our estimates in the three soil depths to predictions of SOC models (Methods section; Fig. 4). To do so, a $Q_{10}$ function (which describes the temperature sensitivity of SOC decomposition) was applied to a one-pool SOC model to predict net steady-state SOC stock changes under different warming levels. The model-predicted changes of SOC stock were calculated as the percentage difference of SOC

between two steady states under ambient and future warmer conditions. Our estimates of temperature responses of net SOC stock changes are well in line with the modelled responses using $Q_{10}$ values of 1.7, 1.4 and 1.2 under a scenario of no carbon input changes in the 0–0.3, 0.3–1 and 1–2 m soil depths, respectively (Fig. 4a). Under scenarios of carbon input changes, $Q_{10}$ values vary depending on the magnitude of carbon input changes. If assuming a 2% increase of carbon inputs per 1 °C warming, $Q_{10}$ values of 2.1, 1.7 and 1.4 are predicted in order to capture the estimated SOC losses in the 0–0.3, 0.3–1 and 1–2 m soil depths, respectively. The decrease of $Q_{10}$ with soil depth may be attributed to decreased substrate availability and increased soil

environmental constraints which inhibit microbial activity and its temperature response[45]. Overall, the estimates using the space-for-time substitution approach can well reflect SOC decomposition and turnover processes formulated by pool-based carbon models (e.g., temperature response functions in most models assume a $Q_{10}$ value of ~2, ref. [44]), and are also in the range observed by field and laboratory experiments[1,21,37]. Our estimates of net SOC changes under warming can provide benchmarks for constraining the temperature sensitivity of SOC decomposition and verifying predictions of soil carbon cycle-climate warming feedbacks by Earth system models.

## Comparison with field warming experiments

We also compared the estimated responses of SOC to warming using our approach with those observed in field warming experiments. Using data from existing meta-analyses[7,26,46–50], we compiled a global data set of a total of 261 observations of $SOC_s$ from 85 field warming experiments including warmed and ambient plots. Across the experiments, temperature in warmed plots are 0.1–7 °C higher than that in ambient plots, and the majority of SOC observations are limited to the top 0.2 m soil. To facilitate comparison, the temperature changes (i.e., warming) were grouped into six categories: <1 °C, 1–2 °C, 2–3 °C, 3–4 °C, 4–5 °C and >5 °C. Meta-analysis using the data (Methods section) shows that SOC in general responds negatively to warming (Supplementary Fig. 7). However, the responses are much weaker compared with the estimates in the top 0.3 m soil layer using our space-for-time substitution approach, particularly under high warming levels (Fig. 2a). Indeed, none of the responses under the six field warming levels are significant ($Q_M = 0.19$, which indicates heterogeneity caused by the six warming categories, $p = 0.66$). This disparity is also common across ecosystem types (Supplementary Fig. 7b). For example, SOC changes in forests are on average neutral in field warming experiments irrespective of warming levels, but are estimated to be significantly negative using our space-for-time substitution approach (Supplementary Fig. 7b).

We propose several reasons for the disparity of net SOC changes under warming between that detected by field warming experiments and that by our space-for-time substitution approach. First, durations of field warming experiments are relatively short ranging from <1 to 25 years with an average of 4.7 years, and thus SOC would be far from steady state. By looking into the relationship between SOC responses to experimental durations, a negative, albeit insignificant, relationship is detected (Supplementary Fig. 8), demonstrating the importance of experimental duration. Second, field manipulation of temperature faces some technical challenges in terms of representing natural gradual climate warming and the relevant plant-soil-microorganism interactions. For example, most warming experiments cannot simultaneously warm the whole soil profile and vegetation canopy (particularly in forests). Third, field warming experiments are still scarce (<20 except in grasslands), and cannot cover the heterogeneous environmental conditions across the globe (Supplementary Fig. 7b). We should use caution when extrapolating results from field warming experiments to infer changes in SOC balance under warming conditions across large extents.

## Digital global mapping of SOC changes under warming

We apply the validated meta-forest model for $SOC_s$ (Fig. 5) to each 1 km grid across the globe using three global maps of SOC stock estimates including WISE[51], HWSD[52] and SoilGrids[53], combining with global digital maps of other environmental predictors (Methods). The three SOC data sets provide critical input (i.e., existing SOC stock which is the most important predictor; Fig. 3a) for the meta-forest model, enabling us to obtain insights into how uncertainties in global SOC stocks influence the prediction of net SOC responses to future warming. Under 2 °C warming, digital global mapping based on the WISE data set indicates that SOC loss mainly occurs in relatively cool regions

(Fig. 5a). Boreal forest will lose >20% of SOC in all soil depths, followed by temperate grassland in the 0–0.3 m soil depth (>20%; Fig. 5b). Despite the global average loss, most tundra and desert soils will accumulate carbon under warming (Fig. 5b, d, f). This result is consistent with predictions by Earth system models[54,55]. However, the prediction uncertainty in tundra and deserts is also larger than in other ecosystems (Supplementary Fig. 9). As the sample size (i.e., soil profiles) of "ambient" and "warm" pairs (<100) is relatively small in tundra systems, new observations would help reduce the uncertainty.

Based on the WISE, HWSD, and SoilGrids data sets, the meta-forest model predicts that, averaging across the globe, SOC stock will decrease by 11.8 ± 5.6%, 7.7 ± 6.2% and 23.2 ± 5.6% in the 0–0.3 m soil, respectively, under 1 °C warming (Supplementary Table 3). In terms of absolute SOC stock changes, using the three data sets, global SOC stock loss is estimated to be 166 ± 46, 115 ± 42 and 343 ± 67 Pg C in the 0–0.3 m soil under 1 °C warming, respectively (Table 1). Aggregating layer-specific absolute losses to the 0–1 m and 0–2 m soil depths indicate that global SOC loss is 620 ± 130 Pg and 1551 ± 293 Pg under 1 °C climate warming, respectively (Table 1). The loss will be increased to 843 ± 176 Pg and 1945 ± 387 Pg in the two depths, respectively, under 5 °C air warming. If assuming 40% of the SOC loss in the 0–1 m soil profile (508 ± 117 Pg based on SOC stock estimates of WISE) will end up in the atmosphere (the remaining 60% would be sequestered to the ocean and terrestrial vegetation) under 2 °C warming, current atmospheric $CO_2$ concentration of ~410 ppm will be increased by ~24% or to ~508 ppm. This result is in line with the close association of temperature with atmospheric $CO_2$ concentration revealed by ice cores[56]. However, it should be highlighted that different estimates of current SOC stocks across the globe result in large uncertainty in the magnitude of SOC loss under future warming (Table 1). The main reason is that there is a large difference of SOC stocks in the three data sets, especially SoilGrids estimates much higher SOC stock in many biomes than HWSD and WISE (Supplementary Fig. 10).

## Limitations and uncertainties

There are several limitations/uncertainties in the space-for-time substitution approach. First, it does not allow us to assess the effects of other global change factors that may accompany with warming, e.g., elevated $CO_2$, nitrogen deposition, and fire. Considering the general positive effects of elevated $CO_2$ and nitrogen deposition on plant growth[57,58], our results would overestimate SOC loss. In terms of fire, warming may lead to more severe and frequent fires, particularly in relatively dry areas[59], altering carbon inputs to soil in terms of both quantity and quality (e.g., more pyrogenic carbon inputs) and physicochemical environment for SOC decomposition[60]. Such fire-induced changes in carbon inputs and outputs and SOC stabilisation processes may interact with warming to regulate SOC balance. Second, we do not consider the potential effects of changes in other climatic variables (e.g., extreme climate events), which may interact with warming to determine the net balance of SOC under warming. Indeed, our results have demonstrated that precipitation seasonality together with other precipitation-related variables are important predictors of the response (Fig. 3). Third, the approach adopts an implicit assumption that the history (e.g., paleoclimate) of the soil profiles experienced in the ambient and warm groups are similar, or has negligible effect on the response of SOC to warming. Although we have partially tested this by only including groups with more than 20 soil profiles to dilute/average the effect of the potential diverse history, the role of soil history related to land use, climate and geological events should be further elucidated.

Assessing a comprehensive observational global data set of whole-soil-profile SOC measurements using an innovative hybrid approach of space-for-time substitution and meta-analysis, we have demonstrated that global soils on average will be a carbon source to the atmosphere under future warmer climate, supporting the

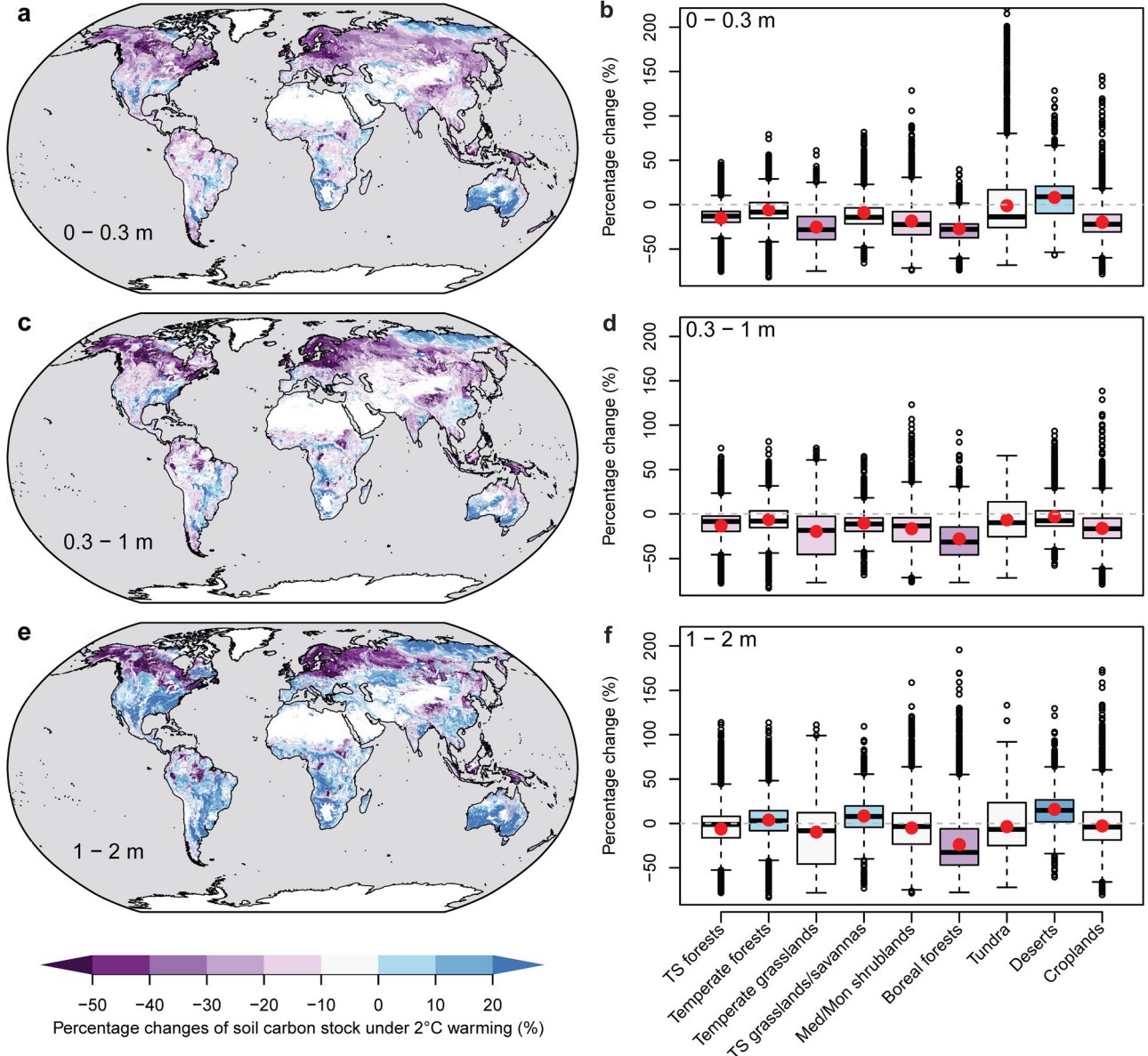

**Fig. 5 | Global pattern of percentage changes of soil organic carbon (SOC) stocks under 2 °C warming.** Top, middle and bottom panels: three soil layer depths (i.e., 0–0.3, 0.3–1, 1–2 m), respectively. Left panels: maps of the spatial distribution of the changes; right panels, aggregated by biome types. In **b**, **d**, **f**, the red points are the average value in each biome; boxplots show the median and interquartile, with whiskers extending to the most extreme data point within 1.5×interquartile range. The maps are produced by applying machine learning-based meta-forest models constrained by the global soil profiles across the globe at the resolution of 1 km. Uncertainties in the mapped changes have been presented in Supplementary Fig. 9. The global SOC map of WISE is used for current standing SOC stocks. TS forests, tropical/subtropical forests; Med/Mon shrublands, Mediterranean/montane shrublands; TS grasslands/savannas, tropical/subtropical grasslands/savannas.

expectation of positive soil carbon loss-climate warming feedbacks[17]. This positive feedback is mainly attributed to net SOC loss in upper soil layers, and carbon in deeper soil depths is less vulnerable to air warming. Overall, this study provides in situ evidence of strong positive soil carbon-climate feedbacks, and estimates a global average SOC loss of ~700 Pg C in the top 1 m soil when soils reach a new steady state under 2 °C warming; and boreal forests are the most vulnerable areas. However, some soils will also sequester carbon under future warming. Particularly, tundra systems and desert will play a vital role in mitigating climate change by sequestering atmospheric carbon under warming. Although the time (which would be decades or even millennia depending on ecosystem types and other environmental conditions) needed to totally manifest this change cannot be quantified using our approach, natural or industrial capture of atmospheric carbon is required to offset such loss which, otherwise, will undermine the efforts of mitigating climate change. The derived global maps of net SOC changes can be used to configure and/or test Earth system models for more robust regional and global implications and reliable predictions, facilitating the development of management strategies for SOC preservation and sequestration under future warming conditions.

## Methods

### WoSIS and permafrost-affected soil profiles

The World Soil Information Service (WoSIS) collates and manages the largest database of explicit soil profile observations across the globe[29]. In this study, we used the quality-assessed and standardised snapshot of 2019 (ISRIC Data Hub). We further screened the snapshot, and

**Table 1 | Global absolute loss of soil organic carbon (SOC) under different warming levels predicted by meta-forest model using SOC stocks from three global maps**

| Data sets | Soil layers (m) | SOC stock (Pg) | The sum of absolute losses across global upland pixels (Pg) | | | | |
|---|---|---|---|---|---|---|---|
| | | | +1 °C | +2 °C | +3 °C | +4 °C | +5 °C |
| WISE | 0–0.3 | 830.3 | −166.1 ± 45.9 | −193.8 ± 51.5 | −213.4 ± 50.1 | −242.7 ± 53.0 | −261.6 ± 64.3 |
| | 0.3–1 | 996.3 | −281.0 ± 61.0 | −313.9 ± 65.2 | −331.6 ± 67.4 | −361.9 ± 73.6 | −379.6 ± 82.6 |
| | 1–2 | 988.2 | −294.5 ± 64.4 | −322.0 ± 69.5 | −335.6 ± 72.0 | −365.6 ± 77.6 | −382.7 ± 84.6 |
| HWSD[a] | 0–0.3 | 708.4 | −115.5 ± 42.2 | −137.0 ± 42.5 | −154.5 ± 43.7 | −180.7 ± 49.6 | −195.7 ± 52.5 |
| | 0.3–1 | 858.0 | −215.6 ± 52.4 | −242.2 ± 55.7 | −257.5 ± 54.8 | −286.1 ± 65.6 | −300.9 ± 70.5 |
| SoilGrids | 0–0.3 | 1172.2 | −343.6 ± 67.2 | −396.3 ± 72.2 | −425.5 ± 74.1 | −462.7 ± 79.1 | −485.0 ± 89.4 |
| | 0.3–1 | 1838.4 | −739.1 ± 120.0 | −817.9 ± 130.6 | −844.8 ± 170.4 | −882.3 ± 152.4 | −906.7 ± 167.5 |
| | 1–2 | 2785.5 | −1278.0 ± 227.0 | −1383.1 ± 203.1 | −1408.0 ± 235.5 | −1444.8 ± 261.5 | −1475.7 ± 285.2 |
| Average across datasets | 0–0.3[b] | 903.6 | −208.4 ± 51.8 | −242.4 ± 55.4 | −264.5 ± 56.0 | −295.4 ± 60.6 | −314.1 ± 68.7 |
| | 0.3–1[b] | 1181.0 | −378.5 ± 74.4 | −422.9 ± 79.4 | −443.6 ± 93.8 | −475.0 ± 91.9 | −494.0 ± 100.9 |
| | 1–2[c] | 1886.8 | −786.3 ± 145.7 | −852.6 ± 136.3 | −871.8 ± 153.8 | −905.2 ± 169.6 | −929.2 ± 184.9 |
| | 0–1[b] | 2134.5 | −620.3 ± 129.6 | −700.4 ± 139.2 | −742.4 ± 153.5 | −805.5 ± 157.8 | −843.2 ± 175.6 |
| | 0–2[c] | 4305.5 | −1551.2 ± 292.8 | −1713.5 ± 296.1 | −1779.5 ± 334.8 | −1880.0 ± 348.6 | −1945.7 ± 386.8 |

Values show the mean ± 95% confidence interval.
[a]HWSD does not report SOC stock in the 1–2 m soil layer depth.
[b]The average values were derived based on SOC stock estimates of WISE, HWSD and SoilGrids.
[c]The average values were derived based on SOC stock estimates of WISE and SoilGrids.

excluded soil profiles with obvious errors (e.g., negative depth values of mineral soil, the value of the depth for the deeper layer is smaller than that of the upper layer). Finally, there is a total of 110,695 profiles with records of SOC content (SOC$_c$, g C kg$^{-1}$ soil) in the fine earth fraction < 2 mm. The soil layer depths are inconsistent between soil profiles. We harmonised SOC$_c$ to three standard depths (i.e., 0–0.3, 0.3–1 and 1–2 m) using mass-preserving splines[61,62], which makes it possible to directly compare among soil profiles. We also calculated SOC stock (SOC$_s$, kg C m$^{-2}$) in each standard depth as:

$$SOC_s = \frac{SOC_c}{100} \cdot D \cdot BD \cdot \left(1 - \frac{G}{100}\right), \qquad (1)$$

where $D$ is the soil depth (i.e., 0.3, 0.7, or 1 m in this study), BD is the bulk density of the fine earth fraction <2 mm (kg m$^{-3}$), and $G$ is the volume percentage gravel content (>2 mm) of soil. Amongst the 110,695 soil profiles, unfortunately, only 18,590 profiles have measurements of both BD and $G$. To utilise and take advantage of all SOC$_c$ measurements, we used generalised boosted regression modelling (GBM) to perform imputation (i.e., filling missing data). As such, SOC$_s$ can be estimated. To do so, for BD and $G$ in each standard soil depth, GBM was developed based on all measurements of that property (e.g., BD) in the 110,695 profiles with other 32 soil properties recorded in the WoSIS database. The detailed approach for missing data imputation has been described in ref. 41.

Together with the WoSIS soil profiles, a total of 2,703 soil profiles with data of SOC$_s$ from permafrost-affected regions were obtained from ref. 30. The original data used in ref. 30 have been obtained, and we used the data of SOC$_s$ in the 0–0.3, 0.3–1, and 1–2 m soil layers in this study. These permafrost-affected profiles compensate for the scarce soil profiles in high latitudinal regions in the WoSIS database. Overall, the soil profiles cover 13 major biome groups although the profile numbers vary among biome types (Supplementary Fig. 1). The profiles also cover various climate conditions across the globe with mean annual temperature (MAT) ranging from −20.0 to 30.7 °C and mean annual precipitation (MAP) ranging from 0 to 6,674 mm.

**Environmental covariates**
MAT and MAP for each soil profile were obtained from the WorldClim version 2 (ref. 63). The WorldClim version 2 calculates biologically

meaningful variables using monthly temperature and precipitation during the period 1970–2000. We obtained global spatial layers of MAT and MAP at the resolution of 30 arcsecond (i.e., 0.0083° which is equivalent to ~1 km at the equator). Soil profiles in the same 0.0083° grid (i.e., ~1 km²) share the same MAT and MAP. Besides MAT and MAP, other climatic variables for each soil profile were also obtained from the WorldClim version 2. The WWF (World Wildlife Fund) map of terrestrial ecoregions of the world (https://www.worldwildlife.org/publications/terrestrial-ecoregions-of-the-world) was used to extract the biome type at each soil profile. The MODIS land cover map[64] at the same resolution of NPP databases was used to identify that if the land is cultivated (i.e., land cover type of croplands and cropland/natural vegetation mosaic) at the location of each soil profile.

**Space-for-time substitution: grouping soil profiles**
We used a hybrid approach of space-for-time substitution and meta-analysis to estimate the response of SOC to warming. Traditionally, space-for-time substitution involves determining regression relationships across gradients at one time[31]. The regression was then used to predict future status under conditions when one or more of the covariates has changed[31]. However, the approach was compromised when the effects of other driving variables such as soil type and landform were not minimised. Regarding SOC dynamics, they would show non-linear relationships[19] with temperature modulated by a series of other environmental covariates (e.g., precipitation, vegetation type).

Based on the idea of space-for-time approach[31], first, we sorted all soil profiles by MAT at the soil-profile locations and designated them into MAT classes with an increment of 1 °C (Fig. 1). Then, we derived pairs of soil profiles, with each pair including a "ambient" and "warm" class (i.e., control vs treatment in meta-analysis language) distinguished by MAT (Fig. 1). The ambient class includes soil profiles with MAT ranging from $i$ to $i + 1$ degree Celsius, where $i$ is the lowest temperature in the class. If 1 °C warming is of interest, for example, the warm class will be identified as the class with MAT ranging from $i + 1$ to $i + 2$ degree Celsius (i.e., one degree higher than that of the ambient class; Fig. 1). To control the effects of precipitation, soil type and topography, soil profiles in both ambient and warm classes were further grouped; and each group must have the same following characteristics:

(1) Landform. A global landform spatial layer was obtained from Global Landform classification - ESDAC - European Commission (europa.eu), and global terrestrial lands were divided into three general landform types: plains including lowlands, plateaus, and mountains including hills.

(2) Soil type. The 12 USDA soil orders were used to distinguish soil types. A global spatial layer of soil orders was obtained from The Twelve Orders of Soil Taxonomy | NRCS Soils (usda.gov). We also independently tested the sensitivity of the results to different soil classification systems by including FAO and WRB soil groups (Soil classification | FAO SOILS PORTAL|Food and Agriculture Organization of the United Nations).

(3) Mean annual precipitation (MAP). MAP cannot be exactly the same between the ambient and warm groups. In practice, we considered that soils meet this criterion if the absolute difference of MAP between ambient and warm soils is less than 50 mm. We also tested the sensitivity of the results to this absolute MAP difference using another value of 25 mm, and found that this difference has negligible effect (Supplementary Fig. 11).

(4) Precipitation seasonality. Precipitation seasonality indicates the temporal distribution of precipitation. In this study, we focused on warming alone, and global warming would also have less effect on this seasonal distribution of precipitation. The seasonal distribution pattern of precipitation was classified into three categories: summer-dominated precipitation, winter-dominated precipitation and uniform precipitation. Precipitation concentration index (PCI) was calculated in R precintcon package to distinguish the three patterns[65]:

$$PCI = \frac{\sum_{i=1}^{12} p_i^2}{\left(\sum_{i=1}^{12} p_i\right)^2} \cdot 100, \qquad (2)$$

where $p_i$ is the precipitation in month $i$ in a particular year. In this study, we used the monthly precipitation from 1970 to 2000 obtained from WorldClim version 2 (ref. 63) to calculate the average $\overline{PCI}$ at the location of each profile. If $\overline{PCI} < 8.3$, precipitation spreads throughout the year (i.e., uniform precipitation). If $\overline{PCI} > 8.3$ and total precipitation from April to September (from October to March in the Southern Hemisphere) is larger than that from October to March (from April to September in the Southern Hemisphere), precipitation mainly occurs in summer (i.e., summer precipitation); otherwise, it is winter precipitation.

By applying these selection criteria to all soil profiles, we obtained pairs (i.e., an "ambient" group vs a "warm" group) of soil profiles mainly distinguished by MAT (i.e., warming). Amongst pairs, they would be different in landform, soil type, MAP and precipitation seasonality, which enables us to address their effects on the response of SOC to warming. We are interested in five warming levels including 1, 2, 3, 4, and 5 °C.

## Meta-analysis: estimation of the response of SOC to warming
Meta-analysis techniques were used to estimate the percentage response of SOC to warming by comparing SOC content and stock in groups in the warm group to that in the ambient group. The log response ratio of soil C (lnRR) to warming for each pair (i.e., an ambient group vs a warm group) of soil profiles was calculated as:

$$lnRR = \ln\left(\frac{\overline{SOC}^*}{\overline{SOC}}\right), \qquad (3)$$

where $\overline{SOC}$ and $\overline{SOC}^*$ are the mean SOC (either content or stock) in groups from ambient and warm class, respectively. In order to provide a robust estimate of global mean response ratio, the individual lnRR values were weighted by the inverse of the sum of within- ($v$) and between-group ($\tau^2$) variances. As such, the global mean response ratio ($\overline{lnRR}$) could be estimated as:

$$\overline{lnRR} = \frac{\sum_i (lnRR_i \times w_i)}{\sum_i w_i}, \qquad (4)$$

where $w_i = \frac{1}{v_i + \tau^2}$ is the weight for the $i^{th}$ lnRR. In addition, we estimated and compared the mean response ratios under different soil orders, landforms, and precipitation concentration patterns. These mean response rates were calculated in weighted, mixed-effects models using the rma.mv function in R package metafor. To assist interpretation, the results of $\overline{lnRR}$ were back-transformed and reported as percentage change under warming, i.e., $(e^{RR} - 1) \times 100$. These back-transformed values were also used for subsequent data analyses.

An implicit assumption underlying the space-for-time substitution approach is that important events or processes which substantially change the succession direction of studied system (e.g., volcano disruption in one class but not in another class, cultivation in one class but not in another class) are independent of space and time (which includes the past and future)[66]. We conducted two sensitivity assessment to test this assumption. First, we repeated all above assessment by excluding soil profiles from croplands since preferential choice of land clearing for cultivation should be common. Second, we repeated all assessment by including only groups having at least 20 soil profiles. This allows the assessed pairs to cover a higher diversity of land history and future land cover/use, diluting the effect of a typical event at a specific soil profile on the estimates.

## Comparison with SOC turnover models
We compared our estimation with predictions by SOC models. A simple one-pool SOC model can be written as:

$$\frac{dC}{dt} = I - k \cdot C, \qquad (5)$$

where $I$ is the amount of carbon input, $k$ is the decay rate of SOC, and $C$ is the stock of SOC. At steady state, $C = I/k$. A $Q_{10}$ function can be applied to estimate $k$ under warming ($k_w$):

$$k_w = k \cdot \exp(0.1 \cdot \triangle T \cdot \log(Q_{10})), \qquad (6)$$

where $\triangle T$ is the warming level. Thus, when soil reaches a new steady state under warming, SOC stock ($C_w$) can be estimated as:

$$C_w = \frac{I_w}{k \cdot \exp(0.1 \cdot \triangle T \cdot \log(Q_{10}))}, \qquad (7)$$

where $I_w$ is the carbon input amount under warming condition. Finally, the response of SOC to warming ($R$) can be calculated as:

$$R = \frac{C_w - C}{C} = \frac{I_w}{I} \cdot \exp(-0.1 \cdot \triangle T \cdot \log(Q_{10})) - 1. \qquad (8)$$

Using Eq. (8), we calculated $R$ under a series of ensembles of $\frac{I_w}{I}$, $\triangle T$, and $Q_{10}$, and compared $R$ with that estimated using our space-for-time substitution approach.

## Comparison with field warming experiments
A number of meta-analyses based on data from field warming experiments had been performed to assess the response of SOC to warming[7,26,46–50], which enable us to conduct comparisons with the estimates using our hybrid approach combining space-for-time substitution and meta-analysis techniques. A total of five meta-analysis papers have been found by searching the Web of Science. We retrieved the response ratios from the identified papers, and compared them to

our estimations. Here, it should be noted that most field warming experiments focused on SOC changes (stock or content) in the top 0.2 m soil layer. We compared them with our estimation of the response of SOC stock in the top 0.3 m soil.

Besides the published results of meta-analysis, we also conducted an independent meta-analysis using data from field warming experiments. The meta-analysis dataset was mainly from published papers on meta-analysis from 2013 to 2020 (see Supplementary Data 1). It should be noted that the field warming experiments manipulate temperature using different approaches such as open/closed-top chamber, infrared radiators and heating cables. For the comparison, we did not explicitly distinguish these approaches. The experimental duration ranged from 0.42 to 25 years with a mean of 4.7 years, and the warming magnitude ranged from 0.1 to 7°C with a mean of 1.92 °C. To ease comparison, field warming levels were classified into 0–1, 1–2, 2–3, 3–4, 4–5, and >5 °C. The same meta-analysis to that assessing soil profile data was used to predict the response ratio of SOC to the above six warming levels. In addition, we divided the data into four ecosystems (i.e., tundra, forest, shrublands and grasslands) and estimated the response ratio in each ecosystem. These estimates based on field warming experiments were compared with those estimated using our space-for-time approach.

### Variable importance and global mapping

We included 15 environmental predictors to derive a meta-forest model, a machine learning-based random forest model adapted for meta-analysis, to map the response of SOC stock/content to warming across the globe at the resolution of 0.0083°. The 15 environmental predictors reflect generally four broad groups of environmental conditions: baseline SOC conditions represented by current standing SOC stock or content, soil order and soil depth; current baseline climatic conditions represented by MAT, MAP, aridity index, precipitation seasonality represented by PCI, the fraction of precipitation in summer, the difference of temperature between ambient and warm groups, the difference of precipitation between ambient and warm groups; topography represented by elevation and landform; and vegetation represented by NPP and biome type.

The metaforest function in the metafor package was used to derive the model. To fit the model, a fivefold cross-validation was conducted. That is, 80% of the derived response ratios was used to train the model, and the remaining 20% to validate the model. The best model hypeparameters were targeted by running the model under a series of parameter combinations, and the model performance was assessed by the rooted mean squared error (RMSE) and determination coefficient ($R^2$). The meta-forest model allows the estimation of the relative influence of each individual variable in predicting the response, i.e. the relative contribution of variables in the model. The relative influence is calculated based on the times a variable selected for splitting when growing a tree, weighted by squared model improvement due to that splitting, and then averaged over all fitted trees which are determined by the algorithm when adding more trees cannot reduce prediction residuals. As such, the larger the relative influence of a variable, the stronger the effect of the variable on the response variable.

Combining with spatial layers of predictors, the meta-forest model for SOC stock was used to predict the response of SOC to warming across the globe at the resolution of 1 km (most data layers are already at the 1 km resolution as abovementioned, for those layers that are not at the target resolution, they were resampled to the 1 km resolution). In the meta-forest model, current standing SOC stock is the most important predictor (Fig. 4). We use three global maps of SOC stocks including WISE[51] (WISE Soil Property Databases | ISRIC), HWSD[52] (Harmonized World Soil Database (HWSD v 1.21) - HWSD - IIASA) and SoilGrids[53] (SoilGrids250m 2.0) to obtain current standing SOC stocks. These three global maps represent the major mapping products of

SOC stock at the global level, and had been widely used for large scale modelling. The derived meta-forest model was applied across the globe to estimate the response ratio of SOC stock in each 1 km pixel. To do so, the same procedure to group the observed soil profiles (Fig. 1) was applied to group global land pixels (section Space-for-time substitution: grouping soil profiles). The only difference is that global mapping uses all pixels instead of the 113,013 soil profiles. In each 1 km pixel, prediction uncertainty was also quantified using estimates of randomly drawn 500 trees of the fitted meta-forest model to calcuate standard deviation of the predictions.

## Data availability

All data used in this study are from publicly accessible data sources. The World Soil Information Service (WoSIS) provides quality-assessed and standardised global soil profile data (ISRIC Data Hub). Three global maps of SOC stocks: WISE[51] was obtain from (WISE Soil Property Databases | ISRIC), HWSD[52] from (Harmonized World Soil Database (HWSD v 1.21) - HWSD - IIASA), and SoilGrids[53] from (SoilGrids250m 2.0). Climate data were obtain from WorldClim version 2 (WorldClim Version2 | WorldClim - Global Climate Data). The global biome type distribution map is available at https://www.worldwildlife.org/publications/terrestrial-ecoregions-of-the-world. The NPP can be accessed at https://ladsweb.modaps.eosdis.nasa.gov/missions-and-measurements/products/MOD17A3. The global landform spatial layer was obtained from Global Landform classification - ESDAC - European Commission (europa.eu). The map of USDA soil orders was obtained from The Twelve Orders of Soil Taxonomy | NRCS Soils (usda.gov). The FAO and WRB soil orders were obtained from Soil classification | FAO SOILS PORTAL | Food and Agriculture Organization of the United Nations, respectively. The ocean boundary data was obtained from http://www.naturalearthdata.com/downloads/.

## Code availability

Code (R scripts) used to assess the data and generate the results[67] are deposited to figshare (https://doi.org/10.6084/m9.figshare.20365551.v1).

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

## Acknowledgements

We thank Niels H. Batjes, Tom Crowther for reading the mansucript and providing helpful comments. Z.L. ackmowledges the funding support from the National Key Research Program of Ministry of Science and Technology of China (grant no. 2021YFE0114500). M.W., X.G., S.Z., L.X., X.M., and L.Z. acknowledge the funding support from the National Natural Science Foundation of China (grant no. 32171639, 41930754). Contributions of U. Mishra were supported through a U.S. Department of Energy grant to the Sandia National Laboratories, which is a multi-mission laboratory managed and operated by National Technology and Engineering Solutions of Sandia, LLC, a wholly owned subsidiary of Honeywell International, Inc., for the U.S. Department of Energy's National Nuclear Security Administration under contract DE-NA-0003525.

## Author contributions

Z.L. conceived the study; M.W. led data assessment with significant contribution of X.G., S.Z., and L.X.; X.G. collected the data from field warming experiments with the contribution of B.Z.; U.M. and Y.Y. contributed to soil profile data; T.Q. and T.J. integrated soil profile datasets; Z.L., M.W., X.G., G.W., and Z.S. interpreted the results with the contribution of all authors, Z.L. wrote the first draft and all authors read and improved the draft.

## Competing interests

The authors declare no competing interests.
