## [Peer Review File · Nature Communications]

Global soil profiles indicate depth-dependent soil carbon losses under a warmer climateREVIEWER COMMENTS

Reviewer #1 (Remarks to the Author):

Line 49-50 Unclear meaning.

Line 51 Do you mean chronological?

Line 52 grammar – need.

Line 62-64 Perhaps more informative to state a broad uncertainty. Certainly, the directional response could be ecosystem-specific.

74-75 I have deep reservations about a space-for-time approach being extrapolated across the globe.

Methods: Only 17% (18,590 of 110,695 profiles) had measured bulk density or gravel measurements, making their calculations of most SOC stock values from assumed values. This is not acceptable.

Space-for-time. I am still very unclear how the warmed and non-warmed pairs were developed.

From the description, there was no control for geography, so warming pairs could be very geographically diverse and confounded within the categorization the authors propose.

Even though the authors classes precipitation, (my understanding was that it was plus or minus 50mm), these differences can be substantial enough to interact with a warming effect.

Line 507 Do you mean combining?

Reviewer #2 (Remarks to the Author):

Review of Wang, Guo, Zhang et al.

In situ global quantification of whole-soil carbon changes under future climate warming
Nature Communications NCOMMS-22-00043-T

I am enthusiastic about this paper, because it addresses a critical feedback to future climate (the response of soils), which has been controversial in the past. Moreover, the paper represents a huge amount of work (an enormous database) but it does so with a refreshing straightforward approach that is easy to follow. The paper suggests rather large global losses of organic matter (carbon) from soils with anticipated global warming. The paper does a nice comparison of their results using in situ measurements (soil profiles) to a smaller number of well-publicized soil-warming experiments, which are not likely to have reported on steady-state conditions.

The response of tundra and Mediterranean ecosystems here is rather intriguing. The former, of course, accumulated carbon during the last continental deglaciation (Harden et al. 1992), so it is perhaps not surprising that they may do so again in the warmer future we anticipate. The effect of fire on the latter is not addressed in this manuscript and probably should be.

I am somewhat confused by the presentation in Table 1, where, for instance, a loss of 1737 PgC from the "global average" soil profile under 2o C warming (last line) is indicated to be a loss of 20.3% (presumably from the content of 4190.4 in the 0-2 m layer. I get 41% for the same calculation. Similarly, I can't make the reported losses in most of the table match the suggested percent losses from the pools indicated in the first column. Can you explain?

The referencing, and methods appear adequate; generally so is the writing, although a careful final edit for English diction and grammar will be needed.

Reviewer #3 (Remarks to the Author):

Wang et al. present a creative analysis on the effect of climate warming on soil C stocks. I really enjoyed reading this manuscript; it takes an original and easy-to-follow approach to analyse an enormous dataset on soil C stocks, and it puts the results in context of current model estimates of

soil C responses to warming.

That said, I have a few quibbles with some of the analyses. Most importantly, I don't find the comparison with manipulative warming experiments very insightful, and potentially even a bit misleading. This is because these experiments only last a few years to decades (4.7 years on average), whereas differences between sites in the place-for-time analysis have developed over centuries to many thousands of years. Soil C stocks in manipulative warming experiments are not at steady state, so cannot be compared directly as is done in this paper. This shortcoming is mentioned in the discussion, but not in the abstract (which currently seems to suggest that manipulative warming experiments underestimate true effects of warming).

On top of that, the last decade has seen numerous syntheses on manipulative warming experiment. As such, the added value of this analysis therefore seems limited. I'd suggest to take these data out of the manuscript entirely, or perhaps move them to the supplementary material.

Secondly, I think the authors need to elaborate on the implications of the fact that the space-for-time analysis deals with soils that are most likely at steady state. The enormous losses that the authors show will not happen overnight, or even over the course of decades. But, some readers might get that impression, given the fact that the warming magnitude categories are linked to climate predictions for the end of this century (e.g. L78-79).

Thirdly, the authors could do a better job explaining the role of environmental variables listed in figure 3 in modulating the warming effect. Apparently, soil C stock is an important predictor for the % change in soil C with warming. This begs the question: what does the relation between soil C stock and % warming effect look like? The authors mention an increase in soil C stocks with warming in tundra soil, which are naturally rich in soil C. So, does this mean that soils that are high in C will generally show a smaller % decrease in soil C stocks with warming, or even an increase? This would be the direct opposite of the results by the Crowther et al. meta-analysis in Nature from a few years ago.

Fourthly, I am a bit confused by the results for Mediterranean shrubland. Whereas the place-for-time analysis suggests a clear decrease in soil C stocks (extended figure 4), the global extrapolation suggests an increase in soil C stocks (figure 4). Can the authors explain this apparent inconsistency?

Fifthly, some of the results in Table 1 seem quite counter-intuitive. For instance, the WISE dataset suggests 880.2 Pg C in the 1-2 soil layer globally. The meta-forest approach predicts that warming by 1C supposedly causes an average decrease of only 0.01% across the globe, but an absolute loss of over 30% of all soil C. How can those two things be true at the same time? This requires additional explanation.

Minor suggestions:

- the authors mention the importance of soil properties in modulating warming effects. In this light, it might be worth including a reference to Hartley et al. (2021), who conducted an analysis somewhat similar to the one presented here, and found that soil clay content was a key factor determining warming effects.
- While the manuscript reads quite well, I spotted several minor grammatical errors. Perhaps the authors could ask someone to have a quick look at their text?

Reference:

Hartley, I. P., Hill, T. C., Chadburn, S. E., & Hugelius, G. (2021). Temperature effects on carbon storage are controlled by soil stabilisation capacities. *Nature communications*, 12(1), 1-7.

Response to comments from the reviewers

Reviewer #1 (Remarks to the Author):

[**Comment 1**] Line 49-50 Unclear meaning.

Response: Thank you for the careful review. The sentence has been revised to: “..., *hindering reliable prediction of net SOC changes at the time scale of SOC turnover which may be centuries or millennia*” (lines 45-46 in the revised manuscript).

[**Comment 2**] Line 51 Do you mean chronological?

Response: Thanks for this question. We have changed “chronical” to “chronic”. (line 47)

[**Comment 3**] Line 52 grammar – need.

Response: Revised accordingly (line 48).

[**Comment 4**] Line 62-64 Perhaps more informative to state a broad uncertainty. Certainly, the directional response could be ecosystem-specific.

Response: Thanks for this suggestion. The sentence has been revised to: “*Indeed, current results on net SOC balance under warming are inconclusive and uncertain depending on experimental conditions, edaphic properties, baseline climate, and ecosystem type* (lines 60-62)”.

[**Comment 5**] 74-75 I have deep reservations about a space-for-time approach being extrapolated across the globe.

Response: Space-for-time substitution is a very widely used approach to explore long-term phenomena in ecology, geography as well as other relevant disciplines¹⁻³. It has been proved that this approach is particularly successful in systems with successional dynamics. Soils are a such system, particularly in the context of chronic climate changes (e.g., warming). Long-term dynamics of soil organic carbon (SOC) under climate warming are very difficult to be quantified in situ due to strong atmosphere-plant-soil interactions which usually need decades or centuries to be detectable or manifest in situ. The space-for-time substitution approach makes this quantification possible. More importantly, we innovatively combined this approach with meta-analysis techniques. In this study, we applied this innovative approach to a massive data set of SOC observations (>100,000 profiles across the globe) to quantify long-term dynamics of SOC under climate warming. Detailed descriptions on the approach and the calculation processes for the quantification can be found in lines 68-84. **For the global mapping, we integrated the findings identified by the hybrid space-for-time substitution and meta-analysis approach to a machine learning-based approach (i.e., meta-forest) adapted for meta-analysis. The uncertainty in the mapping has also been explicitly quantified (lines 500-541).**

We acknowledge that the space-for-time substitution approach has some assumptions/limitations. Sensitivity analyses have been conducted to test the sensitivity of the results to relevant assumptions (e.g., steady state; please refer to lines 84-88, 96-107). In addition, limitations/assumptions and the relevant consequences on the quantification of SOC changes have been explicitly discussed in lines 278-298.

Reference:

- 1 Huang, X., Tang, G., Zhu, T., Ding, H. & Na, J. Space-for-time substitution in geomorphology. *Journal of Geographical Sciences* **29**, 1670-1680, (2019).
- 2 Pickett, S. T. Space-for-Time Substitution as an Alternative to Long-Term Studies in Long-term studies in ecology (edited by G.E. Likens) 110-135 (Springer, 1989).
- 3 Hartley, I. P., Hill, T. C., Chadburn, S. E. & Hugelius, G. Temperature effects on carbon storage are controlled by soil stabilisation capacities. *Nature Communications* **12**, 6713, (2021).

[Comment 6] Methods: Only 17% (18,590 of 110,695 profiles) had measured bulk density or gravel measurements, making their calculations of most SOC stock values from assumed values. This is not acceptable.

Response: Thank the reviewer for pointing out this. It is regretful that only bulk density (BD) and gravel content (G) were directly measured for 17% of the 110,695 soil profiles. We believe the reason is that measuring BD and G is more challenging than measuring SOC content as it needs much more labor, particularly for deep soil layers. However, the 18,590 measurements provide a good data set to derive imputation models to fill the data gap. Indeed, it is a common practice to impute missing BD and G using machine learning- or geostatistical approaches⁴⁻⁸. For example, the well-known Soilgrids data set⁴ used machine learning models to impute BD and G in order to estimate SOC stock. In this study, we also used machine learning-based models to impute missing BD and G values required to calculate SOC stock (please refer to lines 339-346). The detailed approach has been described in a paper using the same WoSIS data we published recently⁵.

In addition, we would like to emphasize that we have also conducted an estimation of the response of SOC to warming using SOC content which is independent of BD and G. The results indicated that percentage responses are similar between the two estimates using SOC content and stock, respectively (Extended Data Fig. 3 and Extended Data Fig. 4 in the revised manuscript).

Reference:

- 4 Poggio, L. *et al.* SoilGrids 2.0: producing soil information for the globe with quantified spatial uncertainty. *Soil* **7**, 217-240, (2021).

- 5 Luo, Z. K., Viscarra-Rossel, R. A. & Qian, T. Similar importance of edaphic and climatic factors for controlling soil organic carbon stocks of the world. *Biogeosciences* **18**, 2063-2073, (2021).
- 6 Ramcharan, A., Hengl, T., Beaudette, D. & Wills, S. A Soil Bulk Density Pedotransfer Function Based on Machine Learning: A Case Study with the NCSS Soil Characterization Database. *Soil Sci Soc Am J* **81**, 1279-1287, (2017).
- 7 Heuvelink, G. B. M. *et al.* Machine learning in space and time for modelling soil organic carbon change. *European Journal of Soil Science* **72**, 1607-1623, (2020).
- 8 Sequeira, C. H., Wills, S. A., Seybold, C. A. & West, L. T. Predicting soil bulk density for incomplete databases. *Geoderma* **213**, 64-73, (2014).

[Comment 7] Space-for-time. I am still very unclear how the warmed and non-warmed pairs were developed. From the description, there was no control for geography, so warming pairs could be very geographically diverse and confounded within the categorization the authors propose.

Response: Thank you for this comment. The description of the warmed and non-warmed (i.e., ambient) pairs has been revised and refined (lines 68-84):

“In this study, we take advantage of a global data set of SOC measurements in 113,013 soil profiles across the globe⁹ which includes 2,703 soil profiles in the northern hemisphere permafrost region¹⁰ (Extended Data Fig. 1) to assess the responses of both SOC content (SOC_c , g C kg⁻¹ soil) and stock (SOC_s , Mg C ha⁻¹) to climate warming, using a hybrid approach combining space-for-time substitution² with meta-analysis techniques (Fig. 1). First, the 113,013 soil profiles were sorted by mean annual temperature (MAT) at the profile locations and divided into classes distinguished by MAT. Depending on the warming level of interest (i.e., 1, 2, 3, 4, 5 °C in this study), an “ambient” and a “warm” class were selected. That is, MAT in the “warm” class must be certain degrees (i.e., 1, 2, 3, 4 or 5 °C) higher than that in the “ambient” class. Considering the potential effects of precipitation including its seasonality, landform and soil type, each class was further divided into groups distinguished by mean annual precipitation, precipitation seasonality, landform and soil type. Then, meta-analysis techniques were applied to the two groups (an “ambient” group vs a “warm” group) that share the same precipitation, landform and soil type to estimate the percentage response of SOC_c as well as SOC_s to warming (i.e., the difference of MAT between the “ambient” and “warm” groups).”

We hope the description is clearer now. To help the reader more easily catch the approach (which indeed is a key novel aspect in this study), we have moved the schematic representation of the approach (previous Extended Data Fig. 2) to the main text (i.e., Fig. 1 in the revised manuscript).

For the control for geography, we have included landform (lowlands, plateaus, mountains) in the selection of warming pairs. That is, warming pairs have the same landform. In addition, elevation has also been considered in the data assessment. In fact,

our approach enables that geography is the same or similar in each non-warmed and warmed pair.

Reference:

- 2 Pickett, S. T. Space-for-Time Substitution as an Alternative to Long-Term Studies in *Long-term studies in ecology* (edited by G.E. Likens) 110-135 (Springer, 1989).
- 9 Batjes, N. H. *et al.* WoSIS: providing standardised soil profile data for the world. *Earth System Science Data* **9**, 1-14, (2017).
- 10 Mishra, U. *et al.* Spatial heterogeneity and environmental predictors of permafrost region soil organic carbon stocks. *Science Advances* **7**, eaz5236, (2021).

[Comment 9] Even though the authors classes precipitation, (my understanding was that it was plus or minus 50mm), these differences can be substantial enough to interact with a warming effect.

Response: Thank you for this comment. From the statistical perspective, similar to geography, the precipitation difference can be considered as a random noise as the warmed and non-warmed pairs used the same criteria to select soil profiles. That is, precipitation in the pairs is not systematically different.

To address the concern on the precipitation difference, we expanded the estimation by constraining the absolute precipitation difference to 25 mm. The results were very similar and did not show significant difference between the two precipitation selection criteria (see Fig. R1 below, Extended Data Fig. 11 in the revised manuscript). The new assessment and results have been added to the manuscript (lines 405-407).

“We also tested the sensitivity of the results to this absolute MAP difference using another value of 25 mm, and found that this difference has negligible effect (Extended Data Fig. 11)”

Fig. R1. Response of soil organic carbon (SOC) stock and content to warming as impacted by two precipitation selection criteria. This figure has also been included in the manuscript as Extended Data Fig. 11.

[Comment 10] Line 507 Do you mean combining?

Response: Thank you for the careful review. Corrected accordingly.

Reviewer #2 (Remarks to the Author):

Review of Wang, Guo, Zhang et al.

In situ global quantification of whole-soil carbon changes under future climate warming
Nature Communications NCOMMS-22-00043-T

[Comment 11] I am enthusiastic about this paper, because it addresses a critical feedback to future climate (the response of soils), which has been controversial in the past. Moreover, the paper represents a huge amount of work (an enormous database) but it does so with a refreshing straightforward approach that is easy to follow. The paper suggests rather large global losses of organic matter (carbon) from soils with anticipated global warming. The paper does a nice comparison of their results using in situ measurements (soil profiles) to a smaller number of well-publicized soil-warming experiments, which are not likely to have reported on steady-state conditions.

Response: Thank you for these positive and encouraging comments.

[Comment 12] The response of tundra and Mediterranean ecosystems here is rather intriguing. The former, of course, accumulated carbon during the last continental deglaciation (Harden et al. 1992), so it is perhaps not surprising that they may do so again in the warmer future we anticipate. The effect of fire on the latter is not addressed in this manuscript and probably should be.

Response: For the response of Mediterranean ecosystems, we note that there was a mistake in the order of biome types presented under the x-axis in previous Fig. 4 and Extend Data Fig. 9. Particularly, the position of Mediterranean/montane shrublands and boreal forest should be and have been exchanged. All relevant results have been revised. We are sorry for the confusion here.

We are grateful for the insightful comment regarding the effect of fire. As this study focuses on warming, our approach implicitly assumes that the intensity and frequency of fire do not change with warming. In this revision, we have expanded the discussion (lines 283-288):

“In terms of fire, warming may lead to more severe and frequent fires, particularly in relatively dry areas¹¹, altering carbon inputs to soil in terms of both quantity and quality (e.g., more pyrogenic carbon inputs) and physicochemical environment for SOC decomposition¹². Such fire-induced changes in carbon inputs and outputs and SOC stabilization processes may interact with warming to regulate SOC balance.”

Reference:

- 12 Pellegrini, A. F. A. *et al.* Fire effects on the persistence of soil organic matter and long-term carbon storage. *Nature Geoscience* **15**, 5-13, (2022).
- 13 Crowther, T. W. *et al.* Quantifying global soil carbon losses in response to warming. *Nature* **540**, 104-108, (2016).

[Comment 13] I am somewhat confused by the presentation in Table 1, where, for instance, a loss of 1737 PgC from the “global average” soil profile under 2o C warming (last line) is indicated to be a loss of 20.3% (presumably from the content of 4190.4 in the 0-2 m layer. I get 41% for the same calculation. Similarly, I can’t make the reported losses in most of the table match the suggested percent losses from the pools indicated in the first column. Can you explain?

Response: Thank you for the careful review. The global average percentage SOC change is **the average of percentage SOC changes in all upland pixels**, but the global absolute SOC loss was the sum of SOC losses which were calculated **as the product of percentage change and SOC stock** in each pixel. For example, if we have two pixels with SOC stocks of 50 and 20 Mg ha⁻¹, respectively, and percentage losses of 20% and 10%, respectively. The average percentage change will be 15%, while the sum of absolute changes will be 12 Mg ha⁻¹ (i.e., 50*20% + 20*10%) which is not equal to 70*15% = 10.5 Mg ha⁻¹. In this revision, we have split the table to two tables (Tables 1 and Extended Data Table 3 in the revised manuscript) and provided more information to explain the table.

[Comment 14] The referencing, and methods appear adequate; generally so is the writing, although a careful final edit for English diction and grammar will be needed.

Response: Thank the reviewer for this positive comment and the suggestion. In this revision, all authors have thoroughly checked the whole manuscript to refine the language.

Reviewer #3 (Remarks to the Author):

[Comment 15] Wang et al. present a creative analysis on the effect of climate warming on soil C stocks. I really enjoyed reading this manuscript; it takes an original and easy-to-follow approach to analyze an enormous dataset on soil C stocks, and it puts the results in context of current model estimates of soil C responses to warming. That said, I have a few quibbles with some of the analyses.

Response: We appreciate these positive and encouraging comments. We have carefully considered each comment and thoroughly revised the manuscript accordingly. Please refer to our point-by-point responses below.

[Comment 16] Most importantly, I don't find the comparison with manipulative warming experiments very insightful, and potentially even a bit misleading. This is because these experiments only last a few years to decades (4.7 years on average), whereas differences between sites in the place-for-time analysis have developed over centuries to many thousands of years. Soil C stocks in manipulative warming experiments are not at steady state, so cannot be compared directly as is done in this paper. This shortcoming is mentioned in the discussion, but not in the abstract (which currently seems to suggest that manipulative warming experiments underestimate true effects of warming).

Response: Thank you for this point. We agree that our estimation cannot be directly compared with warming experiments due to the time scale issue. As manipulative warming experiment is a mainstream approach used to infer SOC dynamics in response to warming, we think the comparison is still informative if we clearly define the preconditions. In this revision, we have clearly clarified that our approach adopts an assumption of steady state in relevant statements/comparisons in the Abstract as well as in the main text and Method section.

[Comment 17] On top of that, the last decade has seen numerous syntheses on manipulative warming experiment. As such, the added value of this analysis therefore seems limited. I'd suggest to take these data out of the manuscript entirely, or perhaps move them to the supplementary material.

Response: Thank you for this suggestion. We have revised and moved the comparison to supplementary materials (Extended Data Figs. 7 in the revised manuscript) as some readers would be interested in this comparison. Indeed, reviewer #2 commented that "The paper does a nice comparison of their results using in situ measurements (soil profiles) to a smaller number of well-publicized soil-warming experiments, ...". In line with warming experiments, in addition, our approach estimated the same trend of more SOC losses under higher warming level albeit the exact magnitudes estimated by the two approaches were substantially different. This can be supportive to reach the general conclusion of negative soil carbon cycle-climate warming feedbacks. For the time scale discrepancies between the two approaches, please refer to our response to **Comment 16**.

[**Comment 18**] Secondly, I think the authors need to elaborate on the implications of the fact that the space-for-time analysis deals with soils that are most likely at steady state. The enormous losses that the authors show will not happen overnight, or even over the course of decades. But, some readers might get that impression, given the fact that the warming magnitude categories are linked to climate predictions for the end of this century (e.g. L78-79).

Response: Thank the reviewer very much for pointing out this. We have clarified these points regarding the time scale issue and revised relevant statements (e.g., lines 227-231, 314-318). The steady state assumption in our approach has been also highlighted (e.g., lines 84-88, 96-107).

[**Comment 19**] Thirdly, the authors could do a better job explaining the role of environmental variables listed in figure 3 in modulating the warming effect. Apparently, soil C stock is an important predictor for the % change in soil C with warming. This begs the question: what does the relation between soil C stock and % warming effect look like? The authors mention an increase in soil C stocks with warming in tundra soil, which are naturally rich in soil C. So, does this mean that soils that are high in C will generally show a smaller % decrease in soil C stocks with warming, or even an increase? This would be the direct opposite of the results by the Crowther et al. meta-analysis in Nature from a few years ago.

Response: Thank the reviewer for this insightful comment. As pointed out by the reviewer, SOC stock is the most important predictor as demonstrated by meta-forest modelling results (please refer to the results presented in Fig. 4). To elucidate the relationship of SOC changes with SOC stocks, we conducted an additional partial dependence analysis (see Fig. R2 below, Extended Data Fig. 6 in the revised manuscript). The result indicates that warming indeed results in more SOC losses in soils with higher SOC stock until reaching a valley. Here, it should be noted that the maximum SOC stock in Crowther et al. (2016) was 125 Mg ha^{-1} , but it is more than $1,500 \text{ Mg ha}^{-1}$ in our study. If we look into the results in the same SOC stock range (i.e., $0 - 125 \text{ Mg ha}^{-1}$), the result is consistent with Crowther et al.¹³.

Fig. R2. Partial dependence of percentage SOC stock changes under 2 °C warming on SOC stock. This figure has also been included in the manuscript as Extended Data Fig. 6.

Here, we would like to note that we found a mistake in the mapping due to that our R scripts did not correctly read the raster layer of SOC (which is an important predictor for the meta-forest model). This resulted in that the global mapping based on this SOC raster layer was not correctly conducted. In this revision, we have re-conducted all mapping work, and the relevant maps and calculations have been revised. In addition, code (R scripts) was provided as a supplementary material.

Reference:

13 Crowther, T. W. *et al.* Quantifying global soil carbon losses in response to warming. *Nature* **540**, 104-108, (2016).

[Comment 20] Fourthly, I am a bit confused by the results for Mediterranean shrubland. Whereas the place-for-time analysis suggests a clear decrease in soil C stocks (extended figure 4), the global extrapolation suggests an increase in soil C stocks (figure 4). Can the authors explain this apparent inconsistency?

Response: Thank you very much for the careful review. We are sorry for the confusion here. There was a mistake here. The order of biome types was wrong. Particularly, the position of Mediterranean/montane shrublands and boreal forest should be exchanged. The relevant figures and description have been corrected. Please also refer to our response to **Comment 12** raised by Reviewer #2. In addition, as we responded to **Comment 19**, we have updated all maps and calculation after correcting a mistake in reading a raster layer of SOC.

[Comment 21] Fifthly, some of the results in Table 1 seem quite counter-intuitive. For instance, the WISE dataset suggests 880.2 Pg C in the 1-2 soil layer globally. The meta-forest approach predicts that warming by 1C supposedly causes an average decrease of only 0.01% across the globe, but an absolute loss of over 30% of all soil C. How can those two things be true at the same time? This requires additional explanation.

Response: Thanks for the careful review. Reviewer #2 had the same comment. Please refer to our response to **Comment 13**.

Minor suggestions:

[Comment 22] - the authors mention the importance of soil properties in modulating warming effects. In this light, it might be worth including a reference to Hartley et al. (2021), who conducted an analysis somewhat similar to the one presented here, and found that soil clay content was a key factor determining warming effects.

Response: Thanks for this recommendation. We have carefully read this paper. It is a very interesting and insightful study, and also closely relevant to our study (e.g., both used the same WoSIS data set). We have cited this paper.

Reference:

Hartley, I. P., Hill, T. C., Chadburn, S. E. & Hugelius, G. Temperature effects on carbon storage are controlled by soil stabilisation capacities. *Nature Communications* 12, 6713, (2021).

[Comment 23] While the manuscript reads quite well, I spotted several minor grammatical errors. Perhaps the authors could ask someone to have a quick look at their text?

Response: Thanks for the careful review. We have carefully checked the whole manuscript for linguistic issues.

REVIEWERS' COMMENTS

Reviewer #2 (Remarks to the Author):

The authors appear to have given careful consideration to my earlier comments and to have revised the manuscript to my satisfaction. I would send it to production.

Reviewer #3 (Remarks to the Author):

This revised version has addressed most of my concerns. A couple of minor points still require attention, though:

- 1) the manuscript still contains several grammatical errors. I pointed out a couple of them in the annotated file, but please note that this list is not complete.
 - 2) L136-138 this text seems to imply that you found soil C losses in Tundra systems. The studies you cite in this section also imply this. However, you found the opposite, i.e. warming increases soil C in these systems!! Thus, this short section needs to be rewritten.
 - 3) L174-175. I appreciate the inclusion of the new supplementary figure. However, Crowther did an analysis on absolute losses in soil C. You conducted an analysis on % losses. Because of the difference in metrics, I don't think these can be compared directly the way you do. For instance, even if % loss would be perfectly stable across all levels of existing soil C (i.e. very different from what you find), one would observe a positive relation between absolute soil C losses under warming and existing soil C stocks. To avoid confusion, I suggest deleting this sentence.
 - 4) L306-308. I don't think this analysis proves that manipulative warming experiments underestimate warming effects. As such, I suggest deleting this statement.
- L238 I think "use caution" makes a bit more sense than "take care" in this sentence.

Finally, it might be worth including a reference to Van Gestel et al. 2018 here. (Nature, 554(7693), E4-E5.), who a) show that previously proposed predictors do a very poor job explaining variations in absolute soil C loss with experimental warming, b) point out the need for experimental data on soil C at losses at lower soil depths, and b) warn against extrapolating results from experimental warming experiments to global levels.

Response to comments from the reviewers

Reviewer #2 (Remarks to the Author):

[Comment 1] The authors appear to have given careful consideration to my earlier comments and to have revised the manuscript to my satisfaction. I would send it to production.

Response: Thank you very much for reviewing our manuscript and providing constructive comments.

Reviewer #3 (Remarks to the Author):

This revised version has addressed most of my concerns. A couple of minor points still require attention, though:

[Comment 2] the manuscript still contains several grammatical errors. I pointed out a couple of them in the annotated file, but please note that this list is not complete.

Response: We are grateful to these suggestions about grammatical errors. We have carefully read through the manuscript to check and correct grammatical issues. All changes have been marked in **BLUE** in the revised manuscript.

[Comment 3] L136-138 this text seems to imply that you found soil C losses in Tundra systems. The studies you cite in this section also imply this. However, you found the opposite, i.e. warming increases soil C in these systems!! Thus, this short section needs to be rewritten.

Response: Thank the reviewer for the careful review. We have revised the statement to: *“In tundra systems, however, contrary to the expectation of negative response of SOC to warming^{17,37,38}...”*.

[Comment 4] L174-175. I appreciate the inclusion of the new supplementary figure. However, Crowther did an analysis on absolute losses in soil C. You conducted an analysis on % losses. Because of the difference in metrics, I don't think these can be compared directly the way you do. For instance, even if % loss would be perfectly stable across all levels of existing soil C (i.e. very different from what you find), one would observe a positive relation between absolute soil C losses under warming and existing soil C stocks. To avoid confusion, I suggest deleting this sentence.

Response: Thanks for this point. Deleted as suggested.

[Comment 5] L306-308. I don't think this analysis proves that manipulative warming experiments underestimate warming effects. As such, I suggest deleting this statement.

Response: As suggested, this statement has been deleted.

[**Comment 6**] L238 I think "use caution" makes a bit more sense than "take care" in this sentence.

Response: Thank you for the careful review. Revised accordingly.

[**Comment 7**] Finally, it might be worth including a reference to Van Gestel et al. 2018 here. (Nature, 554(7693), E4-E5.), who a) show that previously proposed predictors do a very poor job explaining variations in absolute soil C loss with experimental warming, b) point out the need for experimental data on soil C at losses at lower soil depths, and b) warn against extrapolating results from experimental warming experiments to global levels.

Response: Thanks for this suggestion. This is a very interesting paper and relevant to our study. We have cited this paper.